# Sociodemographic Disparities and Parity in Relation to Urinary Incontinence: A Nationwide Primary Healthcare Cohort Study (1997–2018)

**DOI:** 10.3390/jcm11030496

**Published:** 2022-01-19

**Authors:** Christoffer Sundqvist, Xinjun Li, Kristina Sundquist, Filip Jansåker

**Affiliations:** 1Center for Primary Health Care Research, Clinical Research Centre (CRC), Jan Waldenströms Gata 35, Region Skåne University Hospital, 20502 Malmö, Sweden; sundqvist.christoffer@gmail.com (C.S.); xinjun.li@med.lu.se (X.L.); Kristina.sundquist@med.lu.se (K.S.); 2Center for Primary Health Care Research, Department of Clinical Sciences Malmö, Lund University, 20502 Malmö, Sweden; 3Department of Family Medicine and Community Health, Department of Population Health Science and Policy, Icahn School of Medicine at Mount Sinai, New York, NY 10029, USA; 4Center for Community-Based Healthcare Research and Education (CoHRE), Department of Functional Pathology, School of Medicine, Shimane University, Matsue 690-0823, Japan

**Keywords:** epidemiology, immigration, parity, sociodemographic, urinary incontinence

## Abstract

Objectives: Urinary incontinence (UI) is a very common condition in the primary healthcare settings. Few studies have investigated whether sociodemographic factors are related to UI. This nationwide study aimed to investigate whether there is a relationship between sociodemographic factors and UI in women. Methods: A nationwide open cohort study included 2,044,065 women aged 15–50 years. Several national population-based (Sweden) databases including nationwide primary healthcare data were used. The outcome was the time to the first event of any UI diagnosis during the study period (1997–2018). Cox regression models were used to test for associations between individual sociodemographic factors and UI. Results: The study identified 44,250 UI events. These corresponded to 2.16% of the study population and an incidence rate (IR) per 1000 person-years of 1.85 (95% CI 1.84–1.87). In the fully adjusted model, a high age, low education level, and being born outside of Sweden were independently associated with a higher UI risk, while rural living was associated with a lower risk. The income level did not seem to have a large impact. Most notably, women born in the Middle East/North Africa and Latin America/Caribbean had a substantially higher risk of UI with HRs of 2.41 (95% CI 2.33–2.49) and 2.30 (95% CI 2.17–2.43), respectively. Parity was strongly and independently associated with UI. Conclusion: This study presents novel risk factors associated with UI. The findings provide new knowledge concerning the burden of this disease among women, which could be used to provide more equal healthcare for these patients in the future. Previous research allied with these findings suggests using a comprehensive approach targeting health disparities.

## 1. Introduction

Urinary incontinence (UI) is defined as an involuntary loss of urine [1]. Most epidemiological studies have found prevalence rates of around 5–20% [2,3,4,5,6,7]. There are several well-known risk factors associated with UI, for example, age [2,4,5,6,7,8,9,10], parity [3,4,5,7,8,11], obesity [2,3,4,5,7,10], and hysterectomy [3,4,7]. Studies from around the globe have indicated that sociodemographic factors could be related to UI [5,7,8,12], e.g., UI may be more common in women of low socioeconomic status and in white women compared to African American women in the USA. However, no studies have to the best of our knowledge, yet investigated sociodemographic factors in relation to UI on a nationwide basis. This gap in previous research is most likely due to the lack of nationwide population-based datasets derived from primary healthcare data, which is where UI is commonly diagnosed.

Many people around the world do not have access to universal healthcare, which evidently may lead to sociodemographic health inequalities including for UI [5,7,8,12]. Sweden, on the other hand, has universal tax-financed healthcare services, with the goal to provide healthcare on equal terms for the entire population, irrespective of individual sociodemographic status. With this study, we aimed to investigate whether socioeconomic factors (income, education), country of origin, region of residency, and parity affects the risk of being diagnosed with UI in women aged 15–50 in primary healthcare settings.

## 2. Materials and Methods

Design, settings, and study population: An open-cohort study of the risk of a first event of UI in women aged 15–50 years old during the study period was conducted. For the study population, we factored in women identified in Swedish primary healthcare data from 1997 to 2018, giving a total of 2,044,065 women. In addition, the study used individual-level patient data from several other nationwide healthcare and population-based registers. The STROBE statement checklist for cohort studies was considered when conducting the study and writing the manuscript. The study was conducted at the Center for Primary Health Care Research, Department of Clinical Sciences Malmö, Lund University, Sweden.

Outcome variable: The time to the first event of a urinary incontinence (in this paper: UI) diagnosis during the study period was defined as the outcome variable. UI is classified as N39.3 (stress UI) and as N39.4 (other UI) in the 10th revision of the International Classification of Diseases [13]. Outcome events were identified from primary healthcare data (1997–2018). Patients could only be included when having had the outcome once.

Predictor variables: The individual socioeconomic status was defined by the family income and educational status based on the information of the event year in the register, in an identical manner to a recent study [14] of a similar design. Family income was classified into four groups: (i) low (lowest quartile of the study population), (ii) middle low (second quartile), (iii) middle high (third quartile), and (iv) high (highest quartile). Educational level was classified into three categories according to the years of school attendance: (i) compulsory schooling or less (≤9 years), (ii) partly completed high school education (10–11 years), (iii) completed high school and may have progressed to university/college education (≥12 years). In the youngest age group (15–17 years of age), the parents’ (highest) education level was used. Region of residence was based on the information of the event year in the register and divided into three groups: large cities, rural southern, and rural northern Sweden. Country of origin was defined as born in Sweden or originating outside Sweden, i.e., Eastern European countries, Western countries, Middle East/North Africa (MENA), Africa (excluding North Africa), Asia (excluding Middle East) and Oceania, or Latin America and the Caribbean. The variables were defined for each individual included in the open cohort. The definitions of these categories were based on two previous studies on urinary tract infections (UTI) and sociodemographic factors [14,15]. Parity was defined as having given birth to no child (nullipara), one child (unipara), two children, three children, four children, and five or more children.

Data sources: The project involved linkages of several national registries, with creation of relevant datasets for the research aims, data cleaning, and creation of new analytic variables and datasets. All linkages were performed using the unique 10-digit personal identification number that is assigned to each person in Sweden for their lifetime. This identification number was replaced by an encrypted (pseudonymized) version. The registers and data used in this study were the following: nationwide primary healthcare data (study population and outcome), which is a nationwide database that includes diagnoses from primary healthcare consultations from 21 of the 22 regions in Sweden. The national coverage in this register varies based on when the different regions digitalized their healthcare data. In 2015 and at the end of the study period (2018), the register contained 72% [16] and 87% [14] of the national population, respectively. The national Inpatient and Outpatient Registers included hospital discharge diagnoses and diagnoses from outpatient specialist care, respectively [17]. The Medical Birth Register included data on parity [18]. The Cause of Death Register, Population Register and Multi-Generation Registries were used to collect data on deaths, emigration, immigration, incomes, education, and other sociodemographic data. The population registers are nearly 100% complete for the entire national population [19].

Statistical analysis: The time-period started on 1 January 1997 and proceeded until the first event of UI or emigration, death, or end of the study period on 31 December 2018. We used Cox regression models to estimate hazard ratios (HRs) and 95% confidence intervals (CIs), to examine the association between individual-level sociodemographic factors and the time to first event of UI (ICD-10 N39.3 and N39.4) registered during the study period. We used three models in the analysis: Model 1 was a univariate model for each variable, Model 2 was adjusted only for age, and Model 3 was adjusted for age and the individual-level covariates listed above. The proportionality assumptions were controlled and fulfilled by plotting the incidence rates over time and by calculating Schoenfeld (partial) residuals, and these assumptions were fulfilled. Sensitivity analysis was performed including parity as a categorical variable. We used SAS version 9.4 (SAS Institute Inc. Cary, NC, USA) for all statistical analyses and a two-tailed *p*-value of <0.05 was used to define statistical significance in the main outcomes.

Ethical considerations: This study was a non-intervention register study on already collected and pseudonymized secondary data. It was conducted according to the guidelines of the Declaration of Helsinki and approved by the Ethical Review Board in Lund. Access to the used national registries was obtained from the Swedish authorities and all methods were used in accordance with national guidelines and regulations.

Role of funding sources: The funding sources of the study had no role in the study design; the collection, analysis, and interpretation of data; the writing of the report, or in the decision to submit the paper for publication.

## 3. Results

Table 1 describes the age and sociodemographic characteristics of the study population, and the distribution of the number of first events of UI in relation to these factors. The study population consisted of 2,044,065 women aged 15–50 years somewhere during the study period 1997–2018. The proportion of UI was 2.16% (44,250 cases) of the study population during the time period. The majority of the UI occurred after 34 years of age. The 62.6% of the study population living in the larger cities comprised 81.0% of all UI. The 5.7% of Middle Eastern/North African origin represented 10.3% of all UI.

Table 2 shows the incidence rate (IR) per 1000 person-years of UI during the study period, corresponding to an incidence rate per 1000 person-years of 1.85 (95% CI 1.84–1.87). The lowest IR of 0.65 (95% CI 0.63–0.67) was seen in the youngest women (aged 15–24 years). The IRs were highest in the oldest age group of 45–50 years at 3.09 (95% CI 3.04–3.15), women of Middle Eastern/North African origin at 4.04 (95% CI 3.92–4.16), and Latin American/Caribbean origin at 4.20 (95% CI 3.98–4.43). The IRs were similar in the different education and income groups; however, slightly lower IRs were seen in the highest education level and the lowest income level. Women in the larger cities had a higher incidence rate of 2.28 (95% CI 2.26–2.31) compared to 0.99 (95% CI 0.96–1.01) in Southern Sweden. Women originating from outside of Sweden had a higher IR in general.

Table 3 presents the three models. In Model 1, the univariate analysis shows that a low age and living outside of larger cities were associated with lower risks of UI compared to their corresponding references. For example, being a woman aged 15–24 years yielded a HR of 0.21 (95% CI 0.20–0.22) for being diagnosed with UI compared to a woman aged 45–50 years at inclusion. Low education and being from outside of Sweden (especially of Middle Eastern/North African and Latin American/Caribbean origin) were associated with a higher risk of UI compared to their corresponding reference. For example, being a woman of Latin American/Caribbean origin yielded a HR of 2.65 (95% CI 2.50–2.79) compared to Swedish-born women. Women in the lowest income level had a slightly but significantly lower risk of being diagnosed with UI with a HR of 0.93 (95% CI 0.90–0.95). In Model 2, the age-adjusted model yielded similar results for the region of residence. The results for the socioeconomic variables yielded increased risks for the lowest education level and lower income levels. For example, the lowest education level and the lowest income level yielded HRs of 1.32 (95% CI 1.29–1.36) and 1.33 (95% CI 1.30–1.37) compared to high education and a high income, respectively. The risk of being diagnosed with UI for foreign-born women compared to Swedish-born women was mostly weak but increased slightly for Middle Eastern and African women. In Model 3 adjusting for all covariates, most of the significant results for the sociodemographic variables remained but were in general somewhat attenuated. In addition to the lowest education level with a HR of 1.20 (95% CI 1.17–1.23), the middle education level showed a significant increase in the risk of being diagnosed with UI in this model—with a HR of 1.07 (95% CI 1.05–1.10)—compared to the highest education level. Lower income levels remained associated with a slightly increased risk of the first event of UI, although this was not significant for the lowest income level. The increased risks of UI were further reduced in most immigrant groups from the preceding models but remained significantly higher for all foreign groups of women compared to Swedish women. In particular, the Middle Eastern (excluding North African) and Latin American/Caribbean women were found to have an increased risk of UI—with HRs of 2.41 (95% CI 2.33–2.49) and 2.30 (95% CI 2.17–2.43), respectively—compared to Swedish women when adjusting for the other covariates.

In Table 4, the fully adjusted model also includes parity. There was a strong association with parity: the HR of UI was 1.37 (95% CI 1.33–1.41) for one child, which amplified with increasing parity and was highest with a HR of 1.80 (95% CI 1.68–1.94) in women with five or more children. When adjusted for parity, the HRs for foreign-born women increased slightly (except for Eastern European women) compared to Swedish-born women. For example, for the MENA origin, the HR for UI compared to Swedish-born women increased from 2.41 (95% CI 2.33–2.49) to 2.59 (95% CI 2.51–2.68) when adjusting for parity.

## 4. Discussion

This nationwide cohort of 2,044,065 women aged 15–50 years at inclusion showed that 2.16% had a registered diagnosis of urinary incontinence during the study period (1997–2018). All investigated sociodemographic variables and parity were independently associated with the risk of UI. Most notably, a low education level and foreign origin, especially for MENA and Latin American/Caribbean origins, were independently associated with an increased risk of being diagnosed with UI compared to high education and women born in Sweden, respectively. High age, high parity, living in large cities, and to a lesser degree, low income were also associated with UI compared to their corresponding reference groups. Although this study does not examine any causal mechanisms, the findings together with our previous research [14,15] are suggestive of an unequal distribution of urinary tract diseases amongst women in this age group.

To the best of our knowledge, no population-based study on UI on a nationwide basis has been conducted before. Ethnical variations in UI have previously been found in studies conducted outside of Sweden and Europe [3,4,7,12]. Our study found that women of foreign origin had a 24% to 165% higher risk of UI compared to Swedish-born women. In particular, women of MENA (157%) or Latin America/Caribbean (165%) origin had a particularly higher risk compared to women born in Sweden. Even when adjusting for age, socioeconomic factors, and the region of residence, these associations remained, though they were slightly reduced. This is somewhat in contrast to what previous American studies have found, where American white women seem to be at higher risk of developing UI compared to African American women [3,4,7]. Some explanations for this could be that African American women have poorer access to healthcare in the USA compared to minorities in Sweden.

The region of residency was also strongly associated with the risk of the first event of UI, regardless of adjustments. Compared to women living in large cities, women in rural regions had about a 50% lower risk of UI. This might be explained by the longer distance to medical care in the rural parts of Sweden compared to larger cities; a previous study showed that only 25% of women with UI seek care [20] and it is possible that a long distance to medical care might decrease this percentage even further. Women of the lowest and middle education levels were found to have a 7–20% increased risk of UI compared to the highest educated women. However, women in the lowest income level did not have a strong associated higher risk compared to the highest income level. Previous epidemiological studies involving a couple of thousand patients also indicate that a poorer socioeconomic status increases the risk of having UI symptoms [5,7,8]. A possible explanation for this might be that more immigrant women in Sweden generally live under poorer socioeconomic circumstances [21] and in larger cities [22]. Parity was strongly and independently associated with UI. For example, unipara and the highest parity had a 37% and 80% increased risk of UI compared to nullipara, respectively. These findings strengthen the previous evidence on parity as a risk factor for UI [3,4,5,7,8].

Sweden has tax-financed healthcare services that by law should provide care on equal terms for the entire population. Nevertheless, health disparities seem to exist among women [14,23,24,25,26,27,28], indicating that preventive measures may be unequally distributed. This study adds to this indication as it suggests the occurrence of several sociodemographic risk factors associated with UI, which to the best of our knowledge, are beyond the direct medical causality of UI. Instead, multiple explanations for this disparity likely exist and it is possible that synergistic multimorbidity, involving several known risk factors for UI, could be at play. For example, obesity [2,3,4,5,7,10], muscular weakness [12], parity [3,4,5,7,8,29], vaginal delivery-related complications [3,7,10], constipation [12], and comorbidities in general [4,6,7,10] have been associated with increased risk of UI but the association between parity and UI seems to diminish with increasing age [29]. In Sweden, foreign-born women, often living under poor socioeconomic conditions, have indications of higher pregnancy-related complications compared to Swedish-born women [30]. These women, especially from non-European countries, also seem to have a higher hospital admission rate, thus indicating higher levels of comorbidities [25]. A low socioeconomic status, both in Sweden and worldwide, has also been considered to be associated with obesity and health-related comorbidities in women [26,31,32]. Considering this, a synergistic multimorbidity [25,26,30,31,32,33,34] with biological and sociodemographic interactions is a possible explanation for the findings of increased risk of UI, and sociodemographic vulnerability might lead certain women to suffer disproportionately from UI.

As a database study with a stipulated time period, our study risks potential biases on UI history (prior to the first event in the study period) and a missing event not registered as a diagnosis. The study also lacked access to the specific symptoms that concluded in a UI diagnosis. There is also a risk of classification bias and that some patients went undetected as they did not declare that they had such problems. However, it is likely that these potential but important biases are non-differential and thus equally distributed among the groups. It is further important to note that the true incidence of UI was not presented in this study as it was a register study that included a first-event UI and that could be missing patients that do not seek healthcare for UI symptoms. Thus, the rate of UI was expectedly lower than in previous studies (5–20%) [2,3,4,5,6,7]. However, these studies were questionnaire-based and asked for UI symptoms, even though these women had not been actively seeking care for their illness. Considering that women with UI symptoms seldom seek care [20], questionnaire-based studies [2,3,4,5,6,7] evidently yield a higher rate of UI compared to our study. The somewhat lower rate for UI in our study could also partly be explained by the age groups included, which were limited to 15–50 years at inclusion, as a high age is a known risk factor for UI [2,4,5,6,7,8,9,10]. However, the oldest women were aged 72 years at the end of the follow-up and our aim was not to assess the actual incidence of UI in the oldest age groups, but rather risk factors associated with this condition. Furthermore, bearing in mind that only 25% of women with UI seem to seek care [20], the “true” rate of UI during our study period could be somewhere around 8%, which actually correlates well with previous estimations. Additionally, while we had access to parity, we did not have access to some other potential risk factors. That said, the study’s limitations are balanced by its strengths. Major strengths with this study were that it involved several highly validated nationwide databases of good quality [17,19] and minimal missing information, which have recently been used for similar studies on other diseases [14,27,28,35]. Secondly, the overall IR per 1000 person-years was 1.85, with 0.65 in women aged 15–24 years and 3.09 in women aged 45–50 years. This finding demonstrates and strengthens what is generally known: that UI is mainly associated with an older age in women, who over their life course have accumulated known risk factors such as parity, obesity, and other comorbidities associated with UI [2,4,5,6,7,8,9]. Furthermore, compared to previous epidemiological studies on UI, which have mainly been recall surveys and not nationwide studies [2,3,4,5,6,7,8,9,12], this study has the advantage of being a register study and thus diminishing the risk of recall bias and only including diagnosed cases of UI rather than only symptoms of UI, which tend to overestimate the incidence. Previous epidemiological studies have also only involved a couple of thousand patients and lacked the power of including a nationwide study population that involved two million women with over 40,000 events of UI. Finally, this study involved access to primary healthcare data, which is quite unique compared to previous studies, and it is to the best of our knowledge, also the first study on a nationwide basis to investigate sociodemographic differences associated with UI.

The results from this study are similar to the findings of two recently published studies by our group on sociodemographic factors and UTI [14,15], and as UI is a well-known risk factor for UTI [36], the consistencies between these findings were expected. Considering this and that risk factors seem to be similar for UTI and UI, both biological and sociodemographic interactions could exist, thus causing an unequal distribution of urinary tract diseases in women. Hence, a comprehensive approach might be the most suitable to lower the potentially increased disease burden in certain groups. This could include inter alia, targeting common root causes of multimorbid conditions at sociodemographic levels [34]. Examples of this include promoting healthier lifestyles like physical exercise, eating healthier food, and a stronger focus on integrated healthcare and childbirth care for sociodemographic vulnerable groups. More research is, however, needed on possible causal mechanisms explaining the association between sociodemographic factors and UI in this study so that healthcare workers and planners can better understand which interventions to focus on in specific sociodemographic groups.

In conclusion, our findings of increased risks of first-event urinary incontinence in several sociodemographic groups have to our knowledge, not been shown before on a nationwide basis. Altogether, this study provides comprehensive knowledge on the unequal burden of urinary incontinence among women. The findings could help to provide more equal and improved preventive healthcare for this condition in the future. Previous research, together with these findings, suggests that a comprehensive approach targeting the disparities in urinary tract diseases in women is warranted.

## Figures and Tables

**Table 1 jcm-11-00496-t001:** Population and number of events of urinary incontinency in women (1997–2018).

	Total Population		Events of Incontinency	
	No.	%	No.	%
Age group (years)				
15–24	584,389	28.6	4299	9.7
25–34	589,188	28.8	10,895	24.6
34–44	554,795	27.1	17,594	39.8
45–50	315,693	15.5	11,462	25.9
Educational level				
≤9	327,981	16.0	7691	17.4
10–11	424,123	20.8	11,374	25.7
≥12	1,291,961	63.2	25,185	56.9
Family income				
Low	510,155	25.0	9900	22.4
Middle-low	510,580	25.0	12,092	27.3
Middle-high	512,299	25.0	11,894	26.9
High	511,031	25.0	10,364	23.4
Region of residence				
Large cities	1,279,428	62.6	35,863	81.0
Southern Sweden	518,887	25.4	5252	11.9
Northern Sweden	245,750	12.0	3135	7.1
Country of origin				
Sweden	1,563,956	76.5	30,432	68.8
Eastern Europe	115,994	5.7	2966	6.7
Western countries	97,918	4.8	2413	5.5
Middle East/North Africa	115,640	5.7	4571	10.3
Africa (excluding North Africa)	43,068	2.1	736	1.7
Asia (excluding Middle East) and Oceania	78,167	3.8	1799	4.1
Latin America and the Caribbean	29,322	1.4	1333	3.0
All	2,044,065	100.0	44,250	100.0

**Table 2 jcm-11-00496-t002:** Incidence rate (per 1000 person-years) of urinary incontinency in women (1997–2018).

	Incidence Rate (IR), per 1000 Person-Years		
	IR	95% CI	
Age group (years)			
15–24	0.65	0.63	0.67
25–34	1.58	1.55	1.61
35–44	2.65	2.61	2.69
45–50	3.09	3.04	3.15
Educational level			
≤9	2.04	2.00	2.09
10–11	2.09	2.05	2.13
≥12	1.72	1.70	1.74
Family income			
Low	1.74	1.71	1.78
Middle-low	1.86	1.82	1.89
Middle-high	1.92	1.88	1.95
High	1.89	1.86	1.93
Region of residence			
Large cities	2.28	2.26	2.31
Southern Sweden	0.99	0.96	1.01
Northern Sweden	1.10	1.06	1.14
Country of origin			
Sweden (born in)	1.59	1.57	1.61
Eastern Europe	2.51	2.42	2.60
Western countries	2.54	2.44	2.64
Middle East/North Africa	4.04	3.92	4.16
Africa (excluding North Africa)	1.91	1.77	2.05
Asia (excluding Middle East) and Oceania	2.37	2.26	2.48
Latin America and the Caribbean	4.20	3.98	4.43
All	1.85	1.84	1.87

**Table 3 jcm-11-00496-t003:** Association of individual sociodemographic variables and urinary incontinency in women (1997–2018).

	Model 1				Model 2				Model 3			
Covariates	HR	95% CI		*p*-Value	HR	95% CI		*p*-Value	HR	95% CI		*p*-Value
Age (ref. age 45–50 years)												
15–24	0.21	0.20	0.22	<0.0001	0.21	0.20	0.22	<0.0001	0.21	0.20	0.22	<0.0001
25–34	0.51	0.50	0.52	<0.0001	0.51	0.50	0.52	<0.0001	0.48	0.47	0.50	<0.0001
35–44	0.85	0.83	0.87	<0.0001	0.85	0.83	0.87	<0.0001	0.82	0.80	0.84	<0.0001
Educational level (ref. ≥ 12 years)												
≤9	1.20	1.17	1.23	<0.0001	1.32	1.29	1.36	<0.0001	1.20	1.17	1.23	<0.0001
10–11	1.22	1.20	1.25	<0.0001	1.01	0.99	1.03	0.4064	1.07	1.05	1.10	<0.0001
Family income (ref. High)												
Low	0.93	0.90	0.95	<0.0001	1.33	1.30	1.37	<0.0001	1.03	1.00	1.07	0.0361
Middle-low	0.99	0.96	1.01	0.2866	1.14	1.11	1.17	<0.0001	1.08	1.05	1.11	<0.0001
Middle-high	1.02	0.99	1.04	0.2494	1.11	1.08	1.14	<0.0001	1.08	1.06	1.11	<0.0001
Region of residence (ref. Large cities)												
Southern Sweden	0.44	0.43	0.46	<0.0001	0.45	0.44	0.47	<0.0001	0.47	0.45	0.48	<0.0001
Northern Sweden	0.49	0.47	0.51	<0.0001	0.49	0.47	0.51	<0.0001	0.53	0.51	0.55	<0.0001
Country of origin (ref. Born in Sweden)												
Eastern Europe	1.59	1.53	1.65	<0.0001	1.50	1.44	1.55	<0.0001	1.37	1.32	1.42	<0.0001
Western countries	1.61	1.55	1.68	<0.0001	1.30	1.24	1.35	<0.0001	1.23	1.18	1.28	<0.0001
Middle East/North Africa	2.57	2.49	2.65	<0.0001	2.64	2.56	2.72	<0.0001	2.41	2.33	2.49	<0.0001
Africa (excluding North Africa)	1.24	1.15	1.33	<0.0001	1.34	1.25	1.44	<0.0001	1.21	1.12	1.30	<0.0001
Asia (excluding Middle East) and Oceania	1.51	1.44	1.59	<0.0001	1.49	1.42	1.56	<0.0001	1.34	1.28	1.41	<0.0001
Latin America and the Caribbean	2.65	2.50	2.79	<0.0001	2.56	2.43	2.71	<0.0001	2.30	2.17	2.43	<0.0001

Hazard ratio (HR). 95% Confidence interval (CI). Model 1: Univariate model; Model 2: Age-adjusted model; Model 3: Fully adjusted.

**Table 4 jcm-11-00496-t004:** Relationship of parity and sociodemographic variables with urinary incontinency.

Covariates	HR *	95% CI		*p*-Value
Age (ref. age 45–50 years)				
15–24	0.24	0.24	0.25	<0.0001
25–34	0.48	0.47	0.50	<0.0001
35–44	0.82	0.80	0.84	<0.0001
Educational level (ref. ≥ 12 years)				
≤9	1.19	1.16	1.22	<0.0001
10–11	1.05	1.03	1.08	<0.0001
Family income (ref. High)				
Low	0.97	0.94	1.00	0.0478
Middle-low	1.01	0.98	1.04	0.5355
Middle-high	1.04	1.01	1.07	0.0026
Region of residence (ref. Large cities)				
Southern Sweden	0.47	0.45	0.48	<0.0001
Northern Sweden	0.53	0.51	0.55	<0.0001
Country of origin (ref. Born in Sweden)				
Eastern Europe	1.52	1.46	1.58	<0.0001
Western countries	1.28	1.22	1.33	<0.0001
Middle East/North Africa	2.59	2.51	2.68	<0.0001
Africa (excluding North Africa)	1.29	1.20	1.39	<0.0001
Asia (excluding Middle East) and Oceania	1.43	1.37	1.51	<0.0001
Latin America and the Caribbean	2.47	2.34	2.61	<0.0001
Parity (ref. Non)				
One child	1.37	1.33	1.41	<0.0001
Two children	1.39	1.36	1.43	<0.0001
Three children	1.50	1.46	1.55	<0.0001
Four children	1.79	1.71	1.88	<0.0001
Five or more children	1.80	1.68	1.94	<0.0001

Hazard ratio (HR). 95% Confidence interval (CI). * Fully adjusted.

## Data Availability

This study made use of several national registers and owing to legal concerns, data cannot be made openly available. Further information regarding the health registries is available from the Swedish National Board of Health and Welfare: https://www.socialstyrelsen.se/en/statistics-and-data/registers/ (accessed on 12 December 2021) and Kristina Sundquist.

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
