# Peer review of "Sociodemographic Disparities and Parity in Relation to Urinary Incontinence: A Nationwide Primary Healthcare Cohort Study (1997–2018)"

_jcm, 2022, doi:10.3390/jcm11030496_

Round 1

Reviewer 1 Report

The manuscript is valuable research. The evaluation of the evolution of urinary incontinence and risk factors is a topic of interest and the results give a good idea about the influence on public health. However, there are some points that may be reviewed.

First of all, authors include a limitation section. The database and registry used is adequate to evaluate urinary incontinence. The type of incontinence may be affected with age. Please add more data about the types of incontinence and relations with risk factors (overactive bladder, stress incontinence)

Patients include are 15 to 50 years old. It is important that urinary incontinence are more prevalent with age and analysis of patients older than 50 years old would be advisable. Please add a comment as a limitation

Studies evaluating urinary incontinence are available and a commentary is recommended. Publications such as:

Hannestad YS, Rortveit G, Sandvik H, Hunskaar S; Norwegian EPINCONT study. Epidemiology of Incontinence in the County of Nord-Trøndelag. A community-based epidemiological survey of female urinary incontinence: the Norwegian EPINCONT study. Epidemiology of Incontinence in the County of Nord-Trøndelag. J Clin Epidemiol. 2000 Nov;53(11):1150-7. doi: 10.1016/s0895-4356(00)00232-8. PMID: 11106889.

Rortveit G, Hannestad YS, Daltveit AK, Hunskaar S. Age- and type-dependent effects of parity on urinary incontinence: the Norwegian EPINCONT study. Obstet Gynecol. 2001 Dec;98(6):1004-10. doi: 10.1016/s0029-7844(01)01566-6. PMID: 11755545.

Coyne KS, Sexton CC, Thompson CL, Milsom I, Irwin D, Kopp ZS, et al. The prevalence of lower urinary tract symptoms (LUTS) in the USA, the UK and Sweden: results from the Epidemiology of LUTS (EpiLUTS) study. BJU Int. 2009;104(3):352-60.

Andrades M, Paul R, Ambreen A, Dodani S, Dhanani RH, Qidwai W. Distribution of lower urinary tract symptoms (LUTS) in adult women. J Coll Physicians Surg Pak. 2004;14(3):132-5.

Boyle P, Robertson C, Mazzetta C, Keech M, Hobbs FD, Fourcade R, et al. The prevalence of lower urinary tract symptoms in men and women in four centres. The UrEpik study. BJU Int. 2003;92(4):409-14.

Author Response

The manuscript is valuable research. The evaluation of the evolution of urinary incontinence and risk factors is a topic of interest and the results give a good idea about the influence on public health. However, there are some points that may be reviewed.

Response: We appreciate your positive feedback as well as your time and comments in reviewing this manuscript. Please see our responses below to the points raised.

First of all, authors include a limitation section. The database and registry used is adequate to evaluate urinary incontinence. The type of incontinence may be affected with age. Please add more data about the types of incontinence and relations with risk factors (overactive bladder, stress incontinence)

Response: Few previous studies have examined risk factors for UI at the population level and the aim of the present study was not primarily to study different types of UI in the population but rather UI in general; we did therefore not consider different types of UI in this study. We appreciate, however, your suggestion as it could become an interesting future study. Please also note that the oldest individuals were 72 years at the end of the follow-up.

Patients include are 15 to 50 years old. It is important that urinary incontinence are more prevalent with age and analysis of patients older than 50 years old would be advisable. Please add a comment as a limitation

 Response: We agree that UI is more prevalent with higher age, In this study we aimed to follow an open cohort of women aged 15-50 at inclusion in order to explore the risk of an UI diagnose in a 22-year follow-up period. This means that the oldest women were 72 years at the end of the study. New text was, however, included for clarity; please see the limitations in paragraph 5 in the discussion section.

Studies evaluating urinary incontinence are available and a commentary is recommended.

Response: Thank you for these references. These references will help us in our future research, and we have added Rortveit et al. together with a comment in paragraph 4 in the discussion section of the manuscript.

Publications such as:

Hannestad YS, Rortveit G, Sandvik H, Hunskaar S; Norwegian EPINCONT study. Epidemiology of Incontinence in the County of Nord-Trøndelag. A community-based epidemiological survey of female urinary incontinence: the Norwegian EPINCONT study. Epidemiology of Incontinence in the County of Nord-Trøndelag. J Clin Epidemiol. 2000 Nov;53(11):1150-7. doi: 10.1016/s0895-4356(00)00232-8. PMID: 11106889.

Rortveit G, Hannestad YS, Daltveit AK, Hunskaar S. Age- and type-dependent effects of parity on urinary incontinence: the Norwegian EPINCONT study. Obstet Gynecol. 2001 Dec;98(6):1004-10. doi: 10.1016/s0029-7844(01)01566-6. PMID: 11755545.

Coyne KS, Sexton CC, Thompson CL, Milsom I, Irwin D, Kopp ZS, et al. The prevalence of lower urinary tract symptoms (LUTS) in the USA, the UK and Sweden: results from the Epidemiology of LUTS (EpiLUTS) study. BJU Int. 2009;104(3):352-60.

Andrades M, Paul R, Ambreen A, Dodani S, Dhanani RH, Qidwai W. Distribution of lower urinary tract symptoms (LUTS) in adult women. J Coll Physicians Surg Pak. 2004;14(3):132-5.

Boyle P, Robertson C, Mazzetta C, Keech M, Hobbs FD, Fourcade R, et al. The prevalence of lower urinary tract symptoms in men and women in four centres. The UrEpik study. BJU Int. 2003;92(4):409-14.

Reviewer 2 Report

Introduction is good, but it would be good to discuss the differences between different ethnic groups in the introduction as it ultimately informs the discussion

Results are good.

In the discussion the authors should discuss the bias of classification and whether patients may not be registered as having a particular problem unless they declare that they have a problem.

Author Response

Introduction is good, but it would be good to discuss the differences between different ethnic groups in the introduction as it ultimately informs the discussion

Response: We have added new text in the first paragraph of the introduction in accordance with this comment.

Results are good.

Response: Thank you.

In the discussion the authors should discuss the bias of classification and whether patients may not be registered as having a particular problem unless they declare that they have a problem.

Response: Please see the fifth paragraph in the discussion section for a discussion of the different biases. We have also added the potential biases mentioned in your comment.

Reviewer 3 Report

Dear Authors,

your paper provide significant insights on not yet fully explored risk factors associated urinary incontinence. Thank to sound study design and large case number, results are significant. No points need to be furtherly inquired

Author Response

Dear Authors,

your paper provide significant insights on not yet fully explored risk factors associated urinary incontinence. Thank to sound study design and large case number, results are significant. No points need to be furtherly inquired

Response: We are grateful for your positive feedback after the review of this paper.